# Congenital Malformations in a Holstein-Fresian Calf with a Unique Mosaic Karyotype: A Case Report

**DOI:** 10.3390/ani10091615

**Published:** 2020-09-10

**Authors:** Tomasz Uzar, Izabela Szczerbal, Katarzyna Serwanska-Leja, Joanna Nowacka-Woszuk, Maciej Gogulski, Szymon Bugaj, Marek Switonski, Marcin Komosa

**Affiliations:** 1Department of Animal Anatomy, Poznan University of Life Sciences, Wojska Polskiego 71c, 60-625 Poznan, Poland; tomasz.uzar@up.poznan.pl (T.U.); katarzyna.leja@up.poznan.pl (K.S.-L.); 2Department of Genetics and Animal Breeding, Poznan University of Life Sciences, Wolynska 33, 60-637 Poznan, Poland; izabela.szczerbal@up.poznan.pl (I.S.); joanna.nowacka-woszuk@up.poznan.pl (J.N.-W.); sbugaj@tulce.wchirz.pl (S.B.); marek.switonski@up.poznan.pl (M.S.); 3University Centre for Veterinary Medicine, Szydlowska 43, 60-656 Poznan, Poland; maciej.gogulski@up.poznan.pl; 4Department of Preclinical Sciences and Infectious Diseases, University of Life Sciences, Wolynska 33, 60-637 Poznan, Poland

**Keywords:** polyodontia, shortened legs, impaired growth, hypoplastic thymus, chromosome deletion, marker chromosome

## Abstract

**Simple Summary:**

Congenital malformations, defined as defects of morphogenesis present at birth, are an important problem in livestock production, if they are caused by hereditary mutations. They can lead to animal death, reduce their productive ability and influence animal welfare. Thus, the identification of the causes of congenital abnormalities are greatly needed. In the present report, we described a Holstein-Fresian calf with multiple congenital malformations including head asymmetry, the relocation of the frontal sinus and eye orbits, hypoplastic thymus, ductus Botalli, unfinished obliteration in umbilical arteries, and a bilateral series of tooth germs in the temporal region. Cytogenetic examination revealed a unique mosaic karyotype with a small marker chromosome, which could not be identified by standard banding techniques. It can be assumed that the presence the marker chromosome may be associated with observed congenital malformations in the studied calf.

**Abstract:**

A Holstein-Fresian calf with multiple congenital malformations was subjected postmortem to anatomical and genetic investigation. The calf was small (20 kg), had shortened limbs and was unable to stand up. It lived only 44 days. Detailed anatomical investigation revealed the following features: head asymmetry, the relocation of the frontal sinus and eye orbits, hypoplastic thymus without neck part, ductus Botalli, unfinished obliteration in umbilical arteries, and a bilateral series of tooth germs in the temporal region. Cytogenetic examination, performed on in vitro cultured fibroblasts, showed a unique mosaic karyotype with a marker chromosome—60,XX[9 2%]/60,XX,+mar[8%], which was for the first time described in cattle. No other chromosome abnormalities indicating chromosome instabilities, like chromatid breaks or gaps were identified, thus teratogenic agent exposure during pregnancy was excluded. The marker chromosome (mar) was small and it was not possible to identify its origin, however, sequential DAPI/C (4’,6-diamidino-2-phenylindole) band staining revealed a large block of constitutive heterochromatin, which is characteristic for centromeric regions of bovine autosomes. We suppose that the identified marker chromosome was a result of somatic deletion in an autosome and its presence could be responsible for the observed developmental malformations. In spite of the topographic distance among the affected organs, we expected a relationship between anatomical abnormalities. To the of our best knowledge, this is the first case of a mosaic karyotype with a cell line carrying a small marker chromosome described in a malformed calf.

## 1. Introduction

Congenital malformation is defined as a defect of morphogenesis, which developed during intrauterine life, and is observed at birth. These features can range from small anatomical defects to semi-lethal or lethal disorders. Such abnormalities can be caused by genetic (chromosome or gene mutations) or environmental (teratogenic chemical or physical agents, infections, etc.) factors, as well by their combination [1]. They have been identified in a number of livestock species, including bovids [2,3]. Since artificial insemination (AI) is commonly used in cattle breeding, the identification of the gene mutations responsible for congenital malformation is a very important diagnostic issue, which facilitates the eradication of AI bulls carrying a deleterious mutation. In this species, several causative chromosome mutations, such as a trisomy of small autosomes or partial trisomy and monosomy [4,5,6], as well as gene mutations causing skeletal malformation—e.g., complex vertebral malformation or brachyspina [7,8] were identified.

The genetic analysis of a single case of a malformed newborn usually starts with a cytogenetic study focused on searching for chromosome abnormalities. It is known that an abnormal set of autosomal chromosomes (aneuploidies) are responsible for severe, usually lethal, congenital malformations, as it was shown very recently in a stillborn calf with a trisomy of chromosome 29 [5]. Abnormal chromosome complement, manifested by the presence of a small supernumerary marker chromosome, can be also associated with congenital malformation, as it was widely documented in humans [9]. Finally, a cytogenetic analysis can also reveal chromosome instabilities (e.g., chromatid breaks or gaps, micronuclei), which indicate a possible deleterious effect of teratogenic agents during intrauterine life. Such observations were reported in calves with polymelia or amelia [10,11].

In the present report, the anatomical and genetic evaluation of a new case of malformed Holstein-Friesian calf was described.

## 2. Material and Methods

### 2.1. Case Presentation

A Holstein-Friesian calf with congenital malformations, delivered after a pregnancy of normal duration, died at the age of 44 days. Shortened limbs, low body weight (20 kg) and the inability to stand up were the major abnormalities observed by an owner. The owner tried to save the calf, applying supporting feeding and good welfare conditions. The calf was presented postmortem for anatomical and genetical analysis.

### 2.2. Anatomical Analysis

The postmortem detailed investigation was carried out, starting with X-ray examination followed by soft tissue maceration and bone measurements, which did not reveal abnormalities in the number and shape of individual bones in thoracic and pelvic limbs. Measurements of the individual segments of the thoracic and pelvic limbs were compared with analogues in alive calves of the same age and from the same farm.

### 2.3. Cytogenetic Analysis

Chromosome preparations were obtained from in vitro fibroblast culture. The cell culture was established from a skin sample collected postmortem. Cells were routinely cultured using Dulbecco’s modified Eagle medium (DMEM), supplemented with 15% (*v*/*v*) fetal calf serum and 1% (*v*/*v*) penicillin/streptomycin at 37 °C in a humidified atmosphere of 5% CO_2_. The culture medium was changed every 2 days. After 7 days of incubation, when the cells covered the plate, they were trypsinized with 0.25% trypsin solution and then subcultured in T-25 cm^2^ flasks. When 70–80% confluence was reached (24–48 h after seeding), the cells were treated with 0.1 μg/mL Colcemid solution for 2 h. A standard cell harvesting procedure was applied. Chromosomes were analyzed at passage 1 (P1) using Giemsa, DAPI (4’,6-diamidino-2-phenylindole) or sequential DAPI and C-banding techniques. The slides were examined with an epifluorescence Nikon E600 Eclipse microscope equipped with a cooled charge-coupled device CCD digital camera and Lucia software.

### 2.4. Molecular Analysis

Genomic DNA was isolated from skin tissue collected postmortem with the use of the commercial kit (Genomic Mini, A&A Biotechnology, Gdansk, Poland). The polymerase chain reaction (PCR) was applied to detect sex chromosome-linked genes: *SRY* and *AMELY* from the Y and *AMELX* from X chromosomes, respectively. The primers used in the reaction were as follows: for the *SRY* amplification: F—5′ aagaacaacttatgaatagcacca 3′ and R—5′ ttaagtcgcaggtgaaactgt 3′; for the *AMELX* and *AMELY* amplification: F—5′ cagccaaacctccctctgc 3′ and R—5′ cccgcttggtcttgtctgttgc 3′. The PCR conditions were previously described by Szczerbal et al. [11]. The amplicons were analyzed in agarose electrophoresis and their expected sizes were as follows: 851 bp (*SRY*), 280 bp (*AMELX*), and 217 bp (*AMELY*).

## 3. Results

In order to determine the pathological changes, the calf underwent autopsy. External examination indicated a shortening of the limbs. Thus, after X-ray pictures analysis, the bones were submitted to maceration and then measured. Descriptive studies did not indicate abnormalities in the number and shape of individual bones in the thoracic and pelvic limbs. However, as a result of comparative measurements of individual segments of the thoracic and pelvic limbs with analogues in alive calves of the same age and from the same farm, it was stated that the limbs were clearly shortened in all segments (Table 1).

In the next step of the anatomical investigations, several coronary sections of the head in relation to previous X-ray and computed tomography imaging (before maceration) were performed. A preliminary study showed head asymmetry, where the right side was shortened relative to the left one. Computed tomography analyses and anatomical sections showed an uneven skull bone system, which resulted in the relocation of the frontal sinus and eye orbits (Figure 1a,b).

An interesting observation concerned the presence of bilateral series of tooth germs in the temporal region. Both sides of the neurocranium were dominated by dental tissue that invaded the temporal squama as well as pyramid parts. This impaired formation of the internal ear, due to the lack of development of petrous parts of the temporal bone with the cochlear system and semicircular canals. As a result, not only hearing but also a sense of balance could be impaired. Moreover, a large pharyngeal cyst containing several teeth caused pressure on the base of the skull and medulla oblongata (Figure 2a–d).

The topographic location and size of the thorax organs were correct. Only at the heart section did we find a developmental defect. Moreover, the size of the thymus lobe was normal only in mediastinum, while its other parts could not be identified around the trachea in the neck. In healthy calves, the thymus is particularly large compared to other domestic mammal species. It grows especially around the neck, forming two lobes around the trachea, converging into the thorax, which then contacts the thoracic lobe. In the examined calf, these cervical structures were not observed at all, thus this organ was described as hypoplastic (Figure 3a).

The topography of the abdominal and pelvic female organs was preserved. The uterus and external genitals were not altered, however, the ovaries had an elongated shape. Similarly, the features of unfinished obliteration were also observed in umbilical arteries, whose light was still clearly opened (Figure 3b). These vessels fulfilled the role in prenatal period draining blood to the placenta. They should develop over time into round ligaments of the bladder (*ligg. teretia vesicae*). Moreover, a 5 mm patent ductus arteriosus was revealed in the heart.

It is well known that congenital malformations can be caused by an abnormal chromosome complement, thus we decided to perform a cytogenetic study of the affected calf. Only postmortem collected tissue samples were available. Thus, the chromosome preparations were obtained from in vitro cultured fibroblasts. Giemsa or DAPI-stained chromosome preparations revealed a normal chromosome set (60,XX) in 92% (*n* = 102) of the metaphase spreads, while in 8% of metaphases (*n* = 9) a small marker chromosome (60,XX,+mar) was found (Figure 4a–c). Thus, the mosaic karyotype (60,XX/60,XX,+mar) was identified. Unfortunately, it was not possible to identify the origin of this small marker chromosome based on DAPI-banded chromosomes. However, sequential DAPI/C-band staining revealed that the marker chromosome contained a positive C-band (Figure 4d–f). Such a heterochromatic block is characteristic of the centromere of bovine autosomes. Moreover, the analysis of chromosome instability was performed by the assessment of chromosome/chromatid breaks and gaps in 100 metaphase spreads and the presence other micronuclei in 1000 interphase nuclei. No abnormalities indicating chromosome or genome instability were observed (Appendix A). The molecular analysis of genes located on sex chromosomes revealed a lack of the Y-linked (*SRY* and *AMELY*), and the presence of the X-linked *AMELX* genes (data not shown). It was in agreement with the observed set of female XX chromosomes.

## 4. Discussion

A number of anomalies concerning different organs were detected in the studied calf. An important disorder was patent ductus arteriosus (Botalli). This condition has been infrequently reported in calves and the genetic background for this abnormality was suggested by Penrith [12,13]. It was postulated that it can affect body growth and result in poor body condition [14]. An earlier study by West [15] indicated that the cardiac anomalies, including patent ductus arteriosus, are occasionally related with the abnormalities of kidneys, eyes, and the skeletal system. Our study also revealed a shortening of limbs. Interestingly, an association of achondroplasia with patent ductus arteriosus was also reported [16,17].

Anatomical analysis also showed that the inability to stand up after birth was most likely the result of an undeveloped balance organ especially in the left inner ear. The heterotopic polyodontia in this region of the head contributed to this statement. According to the clinical terminology, extra teeth are located outside the normal dental arches. In the examined calf, these abnormalities were associated with the asymmetry of the entire head. The case of ear teeth we described is very similar to the case of horses because in cattle additional teeth are more often found in the maxillary and mandibular regions. Ectopic teeth can lead to the formation of branchiogenic cysts, the removal of which is very difficult because the germs can be strongly connected to surrounding tissues and blood vessels [18,19]. Such dentigerous cysts are developed at the early stages of organogenesis and can be accompanied by a cleft palate and a rather dense hair coat, particularly on the ear pinnae [20]. The only known case of additional teeth located in the ear area was described in Holstein-Friesian heifer by Wapf and Nuss [21]; however, these authors did not report any other concomitant disorders.

Thymus lobe was described as hypoplastic. Since the thymus arises from the walls of the third branch cleft, its right and left buds descend along the neck on both sides of the trachea [17]. Disorders of this process could hypothetically cause the hypoplasia of the thymus. This would coincide with the emergence of ectopic teeth, which also arises as a result of disturbances in the transformation of the branch cleft. Congenital thymic hypoplasia has not been described in detail in animals.

Among different causes of congenital malformations, chromosome abnormalities play an important role. In the studied calf, we observed a very rare mosaic karyotype, including a cell line with a small marker chromosome. The identification of the origin of such markers is not possible on the basis of their morphology and with chromosome banding techniques [22]. Usually, these marker chromosomes occur as additional structures to a normal chromosome complement, called supernumerary marker chromosomes (SMCs), and are identified in mosaic karyotypes. Here, we also detected a mosaic karyotype, including a normal (60,XX) and an abnormal cell line with a small marker chromosome (60,XX,+mar). It should be pointed out, that the marker chromosome was not supernumerary, because the observed number of chromosomes was normal. We did not identify any sings of chromosome instability, which could indicate an exposure of the pregnant cow to teratogens. Thus, we assumed that the observed congenital malformations had a genetic background.

To date, there is no information on the occurrence of a small marker chromosome in domestic animals with congenital malformations. Unfortunately, we were not able to elucidate the origin of this abnormal chromosome. However, a large C-band in the marker chromosome suggests its origin from an autosome. We hypothesized that the zygote had a normal chromosome complement (60,XX) and the occurrence of the cell line carrying the marker chromosome was an effect of a spontaneous somatic deletion. We hypothesized that it was a small deletion, since large deletions usually cause death during fetal development. Small deletions were already reported in cattle. The largest one, identified in chromosome 2 (BTA2) and encompassing 3.7 Mb, was associated with the complete agenesis of horns, facial dysmorphism, growth delay, chronic diarrhea, premature ovarian failure, and variable neurological and cardiac anomalies [23]. Another example is the deletion of 110 kb in BTA18, associated with a lethal underdevelopment, but without visible congenital malformations, of Ayrshire calves [24].

## 5. Conclusions

We observed congenital malformations in the affected calf with mosaic karyotype, probably associated with the presence of an abnormal cell line carrying a small marker chromosome. It is likely that the marker chromosome originated from a somatic deletion in an autosome. To our knowledge, it is the first such case described in cattle.

## Figures and Tables

**Figure 1 animals-10-01615-f001:**
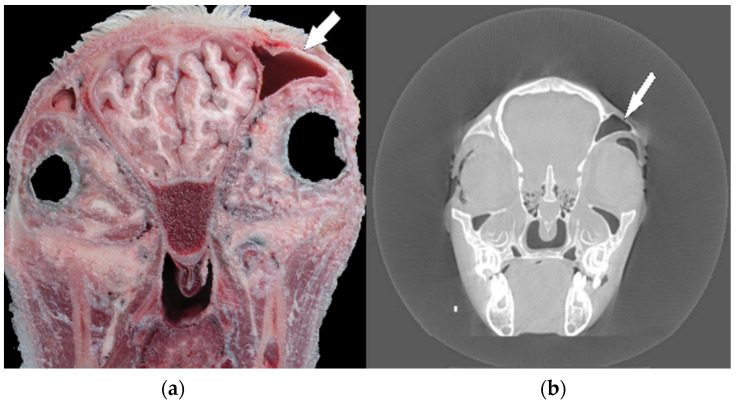
(**a**,**b**) Coronal section on the level of presphenoid bone; view towards rostral. Visible asymmetry of the orbits and frontal sinus (arrow) in the anatomical 4image and computed tomography.

**Figure 2 animals-10-01615-f002:**
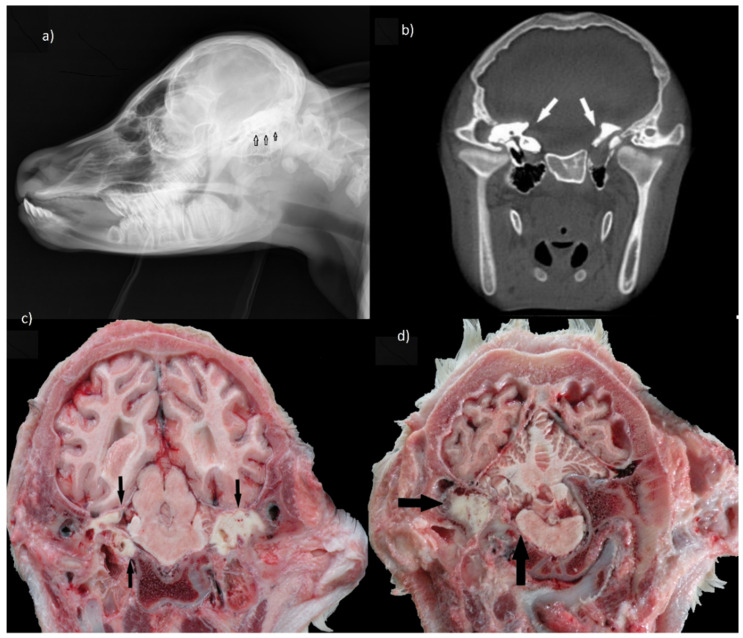
Ear teeth in the rtg picture and anatomical coronal sections; view towards caudal. (**a**) Rtg laterolateral view of the calf ’s head. Tooth-like structures are denser than bone. They are seen in the whole length of the temporal region (arrows). (**b**) The same view (arrows) in the TK coronar section of the temporal region. Asymmetry of the basilar part of the occipital bone and tympanic bulla is observed. (**c**) Coronal section on the level of basisphenoid bone and midbrain (mesencephalon). Dental tissues (arrows) are separated from cerebral hemispheres only by dura mater, without the mediation of osseous tissue. (**d**) Tooth germs (horizontal arrows) interfered with the petrous part of temporal bone. The abnormal osseous structure of this side of the neurocranium exerts pressure on the medulla oblongata causing its deformation (vertical arrow).

**Figure 3 animals-10-01615-f003:**
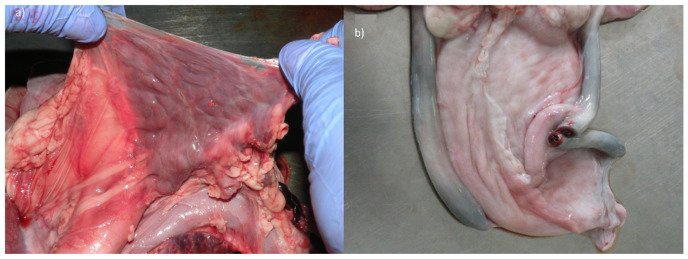
(**a**) The only one part of the thymus in the mediastinum; and (**b**) the unobliterated paired umbilical artery at the urinary bladder. On the right, its cross-sectional view.

**Figure 4 animals-10-01615-f004:**
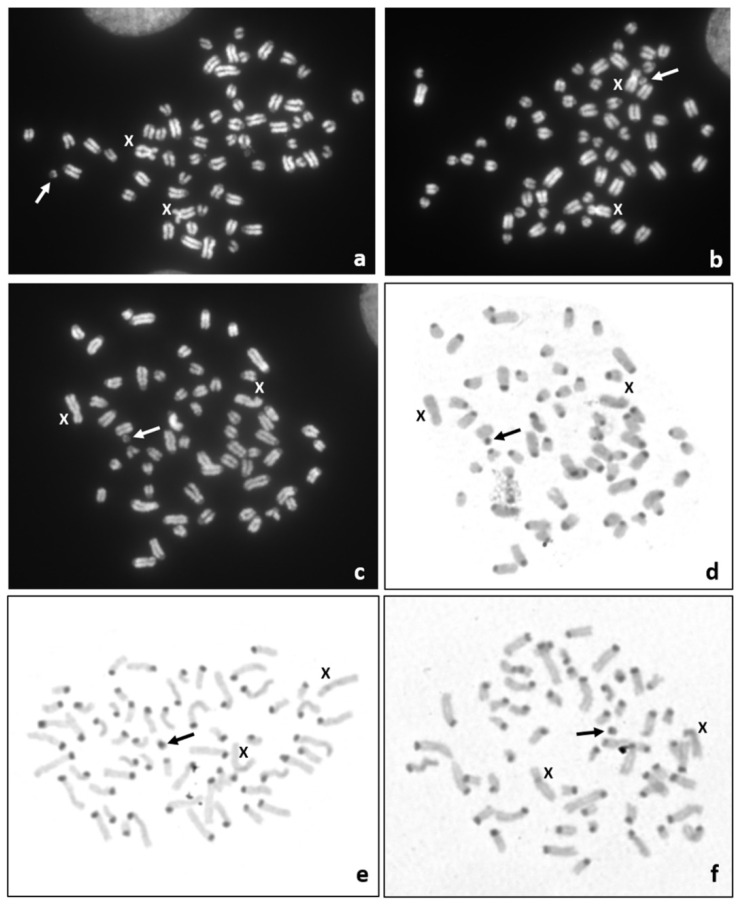
Representative metaphase spreads carrying a small marker chromosome (60,XX+mar): (**a**–**c**)—DAPI staining; (**d**–**f**)—C-banding. Note: sequential DAPI and C-banding of the same metaphase spread (**c**,**d**). The marker chromosome (arrow) and X chromosomes are indicated.

**Table 1 animals-10-01615-t001:** The comparison of thoracic and pelvic limbs in 10 calves in relation to the examined calf (in cm).

Length of Limb Segment	Min	Max	Average	Standard Deviation	Examined Calf
Scapula	19.5	22.0	21.5	0.90	11.5
Humerus	21.0	24.0	22.8	1.14	11.5
Radius and ulna	22.0	26.0	24.2	1.32	14.0
Metacarpus	19.0	24.0	21.8	1.64	12.4
Femur	25.5	28.0	26.7	0.85	13.3
Tibia	24.5	27.5	26.1	1.04	11.7
Metatarsus	28.5	34.5	31.5	1.95	19.4

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
