# Peer review of "Congenital Malformations in a Holstein-Fresian Calf with a Unique Mosaic Karyotype: A Case Report"

_animals, 2020, doi:10.3390/ani10091615_

Round 1
Reviewer 1 Report
Comments on the manuscript Congenital malformations in a Holstein-Fresian calf with a unique mosaic karyotype: A case report
This study manuscript is a case report assessing a calf with phenotypic abnormalities, which was linked to a chromosomal aberration. Since the casuistic in cattle is extremely scarce, this case is quite interesting. But also, the manuscript provides comprehensive data regarding both, phenotype and karyotype.
A couple of important points to correct from my point of view:
L100 In table 1 the authors must analyze the data with a proper statistical methodology to sustain that limbs were shortened in comparison with the other 10 calves.
L150: The authors should state how the absence of breaks and gaps were assessed. Please add a cite at least.
L197. In this paragraph, the authors speculate from an autosomal deletion based on C bandings. That’s correct. But then they speculate with a deletion from a small chromosome, based on a couple of previous cases, which is a little bit risky with the data provided. Maybe that second part could be avoided.
Also, a couple of minor points.
L14 Are the congenital malformations are an important problem in livestock??? I don’t believe that.
L49: I don’t see the relation between IA and congenital malformations (CM). Please support with references or delete.
L78: Please specify the culture time.
L127: Please describe the hearth developmental defect in detail.
Finally, I am not an English native speaker but I believe that the manuscript is fairly well written.
My recommendation is that the manuscript could be published after minor changes.
Author Response
Reviewer #1
This study manuscript is a case report assessing a calf with phenotypic abnormalities, which was linked to a chromosomal aberration. Since the casuistic in cattle is extremely scarce, this case is quite interesting. But also, the manuscript provides comprehensive data regarding both, phenotype and karyotype.
A couple of important points to correct from my point of view:
L100 In table 1 the authors must analyze the data with a proper statistical methodology to sustain that limbs were shortened in comparison with the other 10 calves.
Response: We understand that proper statistics is a key issue in order to demonstrate the problem of shortened legs. However, in statistical methods typically mean values of investigated and control groups are compared. In our research we have dealt with just one case of a calf with shorted legs so we are unable to make a statistical analysis. Table 1s measurements in our case and a comparison with properly developed animals. Thus, we presented only standard deviations and min and max values.
L150: The authors should state how the absence of breaks and gaps were assessed. Please add a cite at least.
Response: The relevant information was added in the text.
L197. In this paragraph, the authors speculate from an autosomal deletion based on C bandings. That’s correct. But then they speculate with a deletion from a small chromosome, based on a couple of previous cases, which is a little bit risky with the data provided. Maybe that second part could be avoided.
Response: We modified the sentence to emphasize that it is our hypothesis only: “We hypothesize that it was a small deletion, since large deletions usually cause death during fetal development.”
Also, a couple of minor points.
L14 Are the congenital malformations are an important problem in livestock??? I don’t believe that.
Response: Thank you for this comment. We modified this sentence by explaining “….. , if they are caused by hereditary mutations.”
L49: I don’t see the relation between IA and congenital malformations (CM). Please support with references or delete.
Response: The sentence was re-written to emphasize the importance of gene mutation diagnostics in AI bulls.
L78: Please specify the culture time.
Response: More details concerning conditions of fibroblast culture were added.
L127: Please describe the hearth developmental defect in detail.
Response: More detailed information about hearh defect was added in the Results section (line 128).
Finally, I am not an English native speaker but I believe that the manuscript is fairly well written.
My recommendation is that the manuscript could be published after minor changes.
Submission Date
12 August 2020
Date of this review
27 Aug 2020 15:02:26
Reviewer 2 Report
Paper titled Congenital malformations in a Holstein-Fresian calf with a unique mosaic karyotype: A case report was presented for review.
The Authors have certainly dedicated a lot of time, work and effort in, anatomical analysis CT/RTG examination and genetic analysis as well as writing the manuscript. The main goal of the case report falls within the scope of the Animals. The manuscript is divided into proper sections.
Keywords
Line 40. In my opinion it should be shortened legs.
Results
Line 100. Table 1 – table column “length of limb segment”. The segments of the limbs are mixed with the names of the bones e.g. femur belongs to skeleton of the femur, brachium this is the term for the segment and is formed by a single bone – humerus.
The measurements were performed after bones maceration. In my opinion it is better to use the name of the bones rather than segments.
Line 108. There is a mistake in Figure 1 caption.
Line 138. It should be ligg. teretia vesicae – the term is used in plural form.
Author Response
Reviewer #2
Comments and Suggestions for Authors
Paper titled Congenital malformations in a Holstein-Fresian calf with a unique mosaic karyotype: A case report was presented for review.
The Authors have certainly dedicated a lot of time, work and effort in, anatomical analysis CT/RTG examination and genetic analysis as well as writing the manuscript. The main goal of the case report falls within the scope of the Animals. The manuscript is divided into proper sections.
Keywords
Line 40. In my opinion it should be shortened legs.
Response: The keyword has been corrected.
Results
Line 100. Table 1 – table column “length of limb segment”. The segments of the limbs are mixed with the names of the bones e.g. femur belongs to skeleton of the femur, brachium this is the term for the segment and is formed by a single bone – humerus. The measurements were performed after bones maceration. In my opinion it is better to use the name of the bones rather than segments.
Response: This mistake has been corrected.
Line 108. There is a mistake in Figure 1 caption.
Response: It has been corrected.
Line 138. It should be ligg. teretia vesicae – the term is used in plural form.
Response: The term has been corrected.
Submission Date
12 August 2020
Date of this review
25 Aug 2020 12:53:45
Reviewer 3 Report
The authors present an interesting case report of developmental abnormalities in a calf, using postmortem morphological analysis and determine the genetic marker responsible for the abnormalities, concluding that it is a mosaic karyotype, with 8% of cellular clones containing a small marker chromosome.
This study would be of interest to readers of Animals journal - however, I notice significant problems with the study design and conclusions, and suggest to substantially revise the paper before considering it for publicaiton.
Major concerns:
1) More details on in vitro cultivation are needed - for how long the fibroblasts were grown? The problem is - possible selective advantage for that particular cellular clone. 8% number is therefore questionable, without proper controls.
2) Related concern - in vitro selection and a new set of conditions (e g use of antibiotics, etc) - could be an explanation for the appearance of small marker chromosome. I agree, it is unusual to see centromeric region in such case, yet in my view entirely possible. I encourage the authors to look at this review by N. Shimizu, which describes similar phenomena: Cytogenet Genome Res 2009;124:312–326 ; PMID: 19556783
3) If the authors suggest the absence of instability related to in vitro selection - more data is needed to support this - table is needed, which would show % of cells without micro nuclei, metaphases without breaks, and so on. Basal level of cells in metaphase that have chromosomal abnormalities is around 1%. and for cells in interphase - 4-5 per 1000 cells. Were the numbers observed similar or different to the above basal levels?
4) One could notice difference in X chromosomes that the authors show - short arm to longer arm length ratio - varies significantly, for example.
Also, ideally, several other calves from the same cow/bull pair need to be examined to compare cytogenetics. Therefore, the authors conclusion that it is the small marker chromosome that is a cause of observed anomalies is unsound. In the revised version I suggest the authors present additional evidence, or tone down the conclusion - not claiming that it is the cause of the observed effect.
Author Response
Reviewer #3
The authors present an interesting case report of developmental abnormalities in a calf, using postmortem morphological analysis and determine the genetic marker responsible for the abnormalities, concluding that it is a mosaic karyotype, with 8% of cellular clones containing a small marker chromosome.
This study would be of interest to readers of Animals journal - however, I notice significant problems with the study design and conclusions, and suggest to substantially revise the paper before considering it for publicaiton.
Major concerns:
1) More details on in vitro cultivation are needed - for how long the fibroblasts were grown? The problem is - possible selective advantage for that particular cellular clone. 8% number is therefore questionable, without proper controls.
Response: The missing information was added.
2) Related concern - in vitro selection and a new set of conditions (e g use of antibiotics, etc) - could be an explanation for the appearance of small marker chromosome. I agree, it is unusual to see centromeric region in such case, yet in my view entirely possible. I encourage the authors to look at this review by N. Shimizu, which describes similar phenomena: Cytogenet Genome Res 2009;124:312–326 ; PMID: 19556783
Response: We wish to explain that we have over 30 year experience in cytogenetic diagnostic studies and we did not observe such marker chromosome in our previous studies as a result of culture condition. Moreover, the same culture conditions we applied at the same time for animals with a normal karyotype. We thank you very much for the suggested reference. Based on this review we could not classify the marker chromosome as double minutes (dmins), which are by definition acentric extra chromosomes. In our study we showed presence of centromeric region (C-band block) in the marker chromosome, what indicate its origin from an autosome.
3) If the authors suggest the absence of instability related to in vitro selection - more data is needed to support this - table is needed, which would show % of cells without micro nuclei, metaphases without breaks, and so on. Basal level of cells in metaphase that have chromosomal abnormalities is around 1%. and for cells in interphase - 4-5 per 1000 cells. Were the numbers observed similar or different to the above basal levels?
Response: Addition analysis was done and an appropriate table (Suppl. Table 1) was added.
4) One could notice difference in X chromosomes that the authors show - short arm to longer arm length ratio - varies significantly, for example.
Response: We measured X chromosomes in 25 metaphases and we found a slight variability of their arm ratio, which oscillated around 2 (see below). We consider the variation as a typical phenomenon associated with a random inactivation of one of two X chromosomes in mammalian females.
Also, ideally, several other calves from the same cow/bull pair need to be examined to compare cytogenetics. Therefore, the authors conclusion that it is the small marker chromosome that is a cause of observed anomalies is unsound. In the revised version I suggest the authors present additional evidence, or tone down the conclusion - not claiming that it is the cause of the observed effect.
Response: Following this suggestion we analyzed a single, unrelated calf as a control and results concerning chromosome instability (micronuclei and chromosome breaks) were added (Suppl.Table 1). Unfortunately, we could not get samples for cytogenetic analysis from half-sibs of the studied case.
Submission Date
12 August 2020
Date of this review
24 Aug 2020 18:43:54
Reviewer 4 Report
In this MS, Tomasz Uzar and colleagues describe a case of congenital malformations in a calf aged 44 days. The case is explored from an anatomical and genetic point of view. This is an interesting study, beautifully illustrated. However, the description of macroscopic, external malformations is minimal, impairing both the reader's understanding of the case clinical presentation and the authors draw adequate conclusions.
During the case description reading, I would think the case to be a duplication of the muzzle of slight to medium importance. Conversely, this hypothesis was not mentioned by the authors in the discussion, focusing mainly on dentigerous cysts and supernumerary teeth. That's why my reading of the clinical condition can be a construct of my mind because of the limited information provided from the description of the calf's gross lesions. Therefore, authors should provide a more detailed description of the lesions and photos of the calf head from different angles, to allow the reader to form an appropriate mental representation of the defect.
Additional concerns are detailed below. Minor issues are included in the attached file.
# 1. the abstract and summary must give the age of the calf
# 2. was the anatomical study restricted to the head? Can the authors provide images from the skull and head's gross anatomy?
# 3. In the Results section and discussion, sometimes we found sentences that represent information to be included in other sections. They represent either a conclusion drawn by the authors from the observed (ex: line 114) or data that was not present in the Results section (e.g., lines 162, 167, and 169).
#4. Figure 2 raised some concerns. They were listed in the image, in the commented copy. Could the areas signaled inside a circle represent a duplication of the muzzle?
#5. The authors refer to supernumerary tooth and dental cysts in the discussion section as a possible close situation to the one they report in the discussion. And conversely, they say nothing about muzzle/head duplication, which is far more frequent in cattle. I am not sure if I can agree with them based on the available information herein. These supernumerary teeth represent a different clinical situation than duplicating a part of the head' structures. The authors should explore both hypotheses and contrast their findings with data obtained from a physical exam (even if postmortem, at the necropsy), other than the anatomical exam presented here. Also, supernumerary teeth are usually found more or less aligned with the normal molar tooth. Was this the case in this calf? Due to the format of cattle head, I would expect to find the dental cysts and/or supernumerary teeth closer to the mandibula - To clarify this and similar issues, a more detailed description of gross malformations is needed.
#6. In their conclusion, the authors establish a causative association between the head malformations and the cytogenetic findings. However, it is not possible with the nature of this study. Authors can only associate these two aspects as being conjointly presented.
#7. The authors refer to a description of the "...surgical/medical management of this case" in the individual contributions list. However, this description was very incomplete (somewhat absent) from the MS. it should be included herein, as already said in this review.
#8. Reference 11 was provided to support the design of the primers used in this MS. However, it does not have such a description/information. Authors should present a table with all the primers used in the study, or provide an alternative reference (ideally in open access)
Minor issues can be found in the attached copy of the MS revised.
Best regards

Author Response
Reviewer #4
Comments and Suggestions for Authors
In this MS, Tomasz Uzar and colleagues describe a case of congenital malformations in a calf aged 44 days. The case is explored from an anatomical and genetic point of view. This is an interesting study, beautifully illustrated. However, the description of macroscopic, external malformations is minimal, impairing both the reader's understanding of the case clinical presentation and the authors draw adequate conclusions.
During the case description reading, I would think the case to be a duplication of the muzzle of slight to medium importance. Conversely, this hypothesis was not mentioned by the authors in the discussion, focusing mainly on dentigerous cysts and supernumerary teeth. That's why my reading of the clinical condition can be a construct of my mind because of the limited information provided from the description of the calf's gross lesions. Therefore, authors should provide a more detailed description of the lesions and photos of the calf head from different angles, to allow the reader to form an appropriate mental representation of the defect.
Additional concerns are detailed below. Minor issues are included in the attached file.
# 1. the abstract and summary must give the age of the calf
Response: The age of the calf was given.
# 2. was the anatomical study restricted to the head? Can the authors provide images from the skull and head's gross anatomy?
Response: No, it was not only the head: all other organs were investigated and all developmental disorders in them are described.
# 3. In the Results section and discussion, sometimes we found sentences that represent information to be included in other sections. They represent either a conclusion drawn by the authors from the observed (ex: line 114) or data that was not present in the Results section (e.g., lines 162, 167, and 169).
Response: It was corrected and included in proper sections.
#4. Figure 2 raised some concerns. They were listed in the image, in the commented copy. Could the areas signaled inside a circle represent a duplication of the muzzle?
Response: No, in the investigated calf the duplication of the muzzle was not observed; however, the asymmetry of the muzzle is present (see figures below).
#5. The authors refer to supernumerary tooth and dental cysts in the discussion section as a possible close situation to the one they report in the discussion. And conversely, they say nothing about muzzle/head duplication, which is far more frequent in cattle. I am not sure if I can agree with them based on the available information herein. These supernumerary teeth represent a different clinical situation than duplicating a part of the head' structures. The authors should explore both hypotheses and contrast their findings with data obtained from a physical exam (even if postmortem, at the necropsy), other than the anatomical exam presented here. Also, supernumerary teeth are usually found more or less aligned with the normal molar tooth. Was this the case in this calf? Due to the format of cattle head, I would expect to find the dental cysts and/or supernumerary teeth closer to the mandibula - To clarify this and similar issues, a more detailed description of gross malformations is needed.
Response: The head asymmetry indicated that tooth were displaced. Some germs were found even in the brain. We demonstrated heteropic polyodontia in the investigated calf, which is defined as an extra tooth, or teeth, outside the dental arcades.
#6. In their conclusion, the authors establish a causative association between the head malformations and the cytogenetic findings. However, it is not possible with the nature of this study. Authors can only associate these two aspects as being conjointly presented.
Response: The conclusion was slightly modified: we used “suppose” instead of “conclude.”
#7. The authors refer to a description of the "...surgical/medical management of this case" in the individual contributions list. However, this description was very incomplete (somewhat absent) from the MS. it should be included herein, as already said in this review.
Response: The missing information was added in the MS.
#8. Reference 11 was provided to support the design of the primers used in this MS. However, it does not have such a description/information. Authors should present a table with all the primers used in the study, or provide an alternative reference (ideally in open access).
Response: The missing details concerning molecular analysis (the name of the commercial kit used for DNA isolation, as well as PCR primer sequences) were added. Moreover, a proper reference (Szczerbal et al., 2014) on this information was added in the list of references (position 11).
Minor issues can be found in the attached copy of the MS revised.
Best regards
Submission Date
12 August 2020
Date of this review
26 Aug 2020 12:56:27
Round 2
Reviewer 3 Report
The supplementary Table S1 added, I am fully satisfied with the authors' response to my concerns and recommend to publish the article in its present form.
Author Response
Dear reviewer,
thank you for all your correction.
Reviewer 4 Report
IN this revised version of the MS, the authors attended most of my concerns except one: to provide a photo from the head of the calf (external view). In my opinion, that image would be important for readers to fully understand the reported clinical case and the authors´ conclusions. It would particularly allow discarding the hypothesis of a duplicated structure in the calf head, which is a more common clinical situation than the one described here.
Some minor issues or suggestions were included in the commented MS, in attach

Author Response
Thank you for all your correction.
We improved our manuscript according to all your suggestions. The terms you indicated are added. We also added a supplementary material – the photo of the investigated calf.